# Feral Animal Populations: Separating Threats from Opportunities

Eduardo J. Rodríguez-Rodríguez [1],*, Jesús Gil-Morión [2] and Juan J. Negro [3]

1 Departamento de Ciencias Integradas, Universidad de Huelva, Campus el Carmen, 21007 Huelva, Spain
2 Sociedad Gaditana de Historia Natural, C/Madreselva s/n, 11408 Jerez de la Frontera, Spain
3 Department of Evolutionary Ecology, Estación Biológica de Doñana-CSIC, Avda. Americo Vespucio 26, 41092 Sevilla, Spain
* Correspondence: eduardo.rodriguez@dci.uhu.es

**Abstract:** Feral animals are those that live in the wild but are descendants of domesticated populations. Although, in many cases, these feral populations imply a demonstrable risk to the ecosystems in which they live and may conflict with local wild species and human activities, there are feral populations that are considered worth preserving and, in some cases, they already enjoy protection by interest groups and even public authorities. In this review, we aim to identify valuable populations using three criteria: (a) Genetic conservation value (for instance, if the wild ancestor is extinct), (b) the niche occupancy criterion and, finally, (c) a cultural criterion. We propose a detailed analysis of feral populations under scrutiny, supporting control measures when necessary, but also allowing for international protection at the same level as wild animals for feral taxa of special concern. Feral taxa, which are already in the focus of conservation efforts, and should be awarded extended recognition and protection, mainly include ancient lineages with relevant genetic or cultural importance.

**Keywords:** domestication; conservation; biodiversity; impacts on native biodiversity; restoration; rewilding

## 1. Introduction

Feralization is a process by which a domestic population becomes free living in the wild. Therefore, this phenomenon might be interpreted as the reversion of domestication, and it implies changes in behavior and phenotype and, mediated by natural selection [1], may even cause gene modifications in the long term [2]. The origin of the ancient domestications in the Neolithic, when all first domesticated species co-existed with their wild counterparts, is in itself an enigmatic anomaly within the evolution of the human lineage [3]. Here, we will specifically refer to the major domestications of livestock and pets while excluding tamed animals, such as falconry birds or cheetah, used for game hunting for millennia and that occasionally escape to the wild with a still intact genetic makeup.

Domestic animals have become feral in many areas worldwide but at different times and circumstances, sometimes soon after the onset of domestications in the mid Holocene and more often quite recently in the Anthropocene [4,5]. In fact, feralization is an ongoing process due to the continual release or escape of domestic animals into the wild [5,6]. Many feral populations cause a severe impact on native wild species, like domestic cats (*Felis silvestris*) predating on songbirds and other fauna [7–11], or goats (*Capra aegagrus hircus*) feeding on endemic flora in some islands like Galapagos [12]. However, many populations, even of species that generate a conservation problem in certain places, such as the above-mentioned goats, may hold conservation value if, for instance, they fill the empty niche of an extinct species, represent the only surviving genetic material of extinct ancestors, or have strong cultural importance.

Currently, the policies regarding feral populations are heterogeneous and sometimes unclear from an international point of view. While wild species are protected by international agreements, such as CITES (Convention on International Trade in Endangered

Species of Wildlife and Fauna) regulating trade [13], or conservation status lists, like the IUCN Red List [14], in addition to the local laws, there is no such coverage for feral animals. However, there are exceptions. Feral horses (*Equus ferus caballus*) occurring in the Herd Management Areas established by the United States Bureau of Land Management are awarded some protection [15]. The Balearic wild goat, a feral goat living in mountain areas on Majorca Island (Spain) [16], is listed in the European Mammal Assessment (listing 260 mammalian species following IUCN criteria).

More commonly, feral animals are included in catalogs of invasive species, promoting their eradication to preserve native wildlife. However, these legislations are greatly heterogeneous among territories and species, and many, if not most of these feral populations, remain unregulated or with non-effective management. The management, if any, is usually assigned to agricultural agencies rather than the ones in charge of wildlife. In addition, basic and applied research on the effect of feral populations in ecosystems is scant, often due to politicization and public pressure [17], with science and scientists playing a secondary role. Our aim is to provide arguments to defend the potential conservation value of some feral populations that may need and deserve legal protection, as with wild species covered by international treaties, such as CITES and the Red List of the International Union for Conservation of Nature (IUCN). After all, artificial biodiversity created by humans stems directly from wild biodiversity at the onset of domestication [18], often as a result of fast selective sweeps resulting in the fixation of desired traits [19].

We propose a detailed assessment of each case based on biological and cultural aspects in order to have several criteria at hand to decide if particular feral populations should be eradicated, controlled, or protected nationally or internationally (e.g., if included in the CITES Appendices or in the IUCN assessments).

## 2. Not All Ferals Are Equal

More often than not feral populations pose an undeniable threat to biodiversity and human activities, such as agriculture and livestock production, and need to be managed or controlled [20]. A dramatic example is provided by feral cats (*F. silvestris catus*), that prey on birds, reptiles, amphibians, and small mammals [7–11], having driven some to extinction, as with the island endemic and flightless Stephens Island wren (*Traversia lyalli*) in New Zealand [21]. In addition, free-roaming domestic cats suppose a risk of disease transmission to threatened fauna [22] or hybridization with genuine wildcats *F. silvestris* [23]. Feral dogs may also cause damage, including subtle effects on endangered predators that are displaced [24]. Another example is the case of feral goats on many islands, including biodiversity sanctuaries, such as the Galapagos Islands [12]. Feral pigs (*Sus scrofa domestica*) also create conservation problems around the world, with conspicuous examples in the New World [25,26]. Another textbook example of a species becoming a pest is the feral rabbit (*Oryctolagus cuniculus*), the only rabbit species that has been domesticated so far [27] and that has been purposely introduced in different parts of the world or has escaped from farms and homes [28]. A case in point among birds is that of chicken feralization (*Gallus gallus*) in places such as Hawaii or Easter Island [29]. The rock dove or common pigeon (*Columba livia domestica*), currently the most cosmopolitan bird via worldwide introductions, may not compete directly with wild species due to its urban living, but it may become an epidemiological concern and provoke damage to human property [30]. Pigeon control is now an industry in its own right, particularly in monumental cities with protected monuments [31]. There are even feral fishes with reported negative impacts, including populations of the royal morph of carps (*Cyprinus carpio*) or domestic morphs of *Carassius auratus* (e.g., the red morph). These populations are highly invasive and count among the main reasons for the alarming conservation status of numerous native freshwater fishes around the world [32]. In addition, some interactions that may lead to behavioral changes in some local wildlife species [33] with unforeseen consequences need to be assessed thoroughly.

However, there are some cases of feral animal populations that may hold conservation interest, based on several criteria that we review here (see Table 1 and Figure 1). The selected criteria rest on three aspects we consider essential in the biological conservation scheme: The conservation of genetic material [34], the preservation of ecological roles [35], and, finally, its importance as cultural heritage [36]. It is important to remark that we only aim to provide an overview of feral taxa with conservation interests, differentiating them from those that imply a threat to biodiversity, and thus, we do not discuss the methods and the animal welfare scheme that must arise once a feral population becomes a risk to biodiversity. However, there is growing support worldwide for the avoidance of lethal or unethical treatments in the management of any animal, wild or domesticated (see, e.g., the rise of veganism), this being in itself a possible new criterion for the abolition of eradication practices.

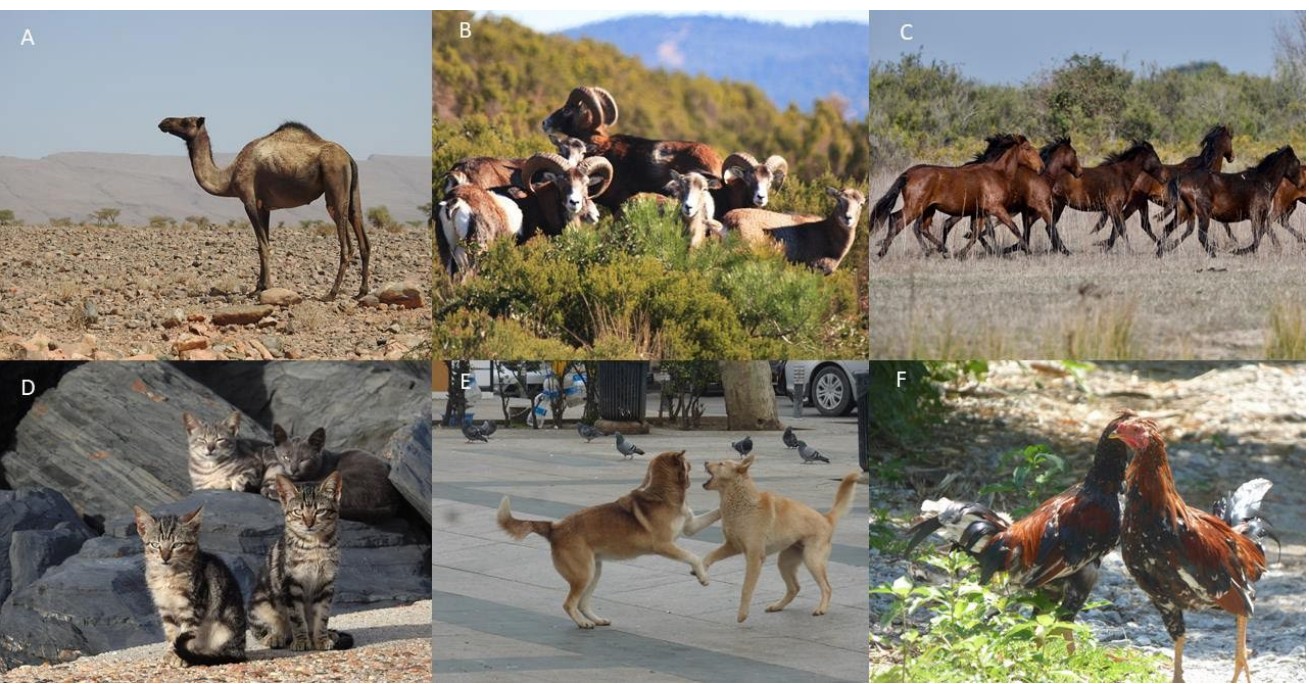

**Figure 1.** Examples of feral animals considered of conservation interest (**A**–**C**) or as a risk for biodiversity (**D**–**F**). Note that the consideration of interest or risk applies only to particular areas. (**A**) Dromedary (*Camelus dromedarius*) in Sahara Desert (Morocco). (**B**) Group of male and female Corsican mouflons (*Ovis orientalis musimon*) introduced to Cádiz mountains (Spain). We only consider of conservation interest the populations in Corsica and Sardinia, their places of origin. (**C**) Feral horse (*Equus ferus caballus*) of the Retuertas breed in Doñana National Park (Spain). (**D**) Feral cat family (*Felis catus*) in Huelva (Spain). (**E**) Feral dogs (*Canis lupus familiaris*) fight and feral pigeons (*Columba livia domestica*) in Istanbul (Turkey). (**F**) Feral chickens (*Gallus gallus domesticus*) in Bac Lieu (Vietnam). (**A**,**D**–**F**): Rodríguez-Rodríguez, E.J.; (**B**,**C**): Negro, J.J.

## 3. Feral Populations Playing Positive Roles

First, some populations may have a reduced impact or even fulfill an important role in ecosystems if they happen to exploit an empty niche. Although the occupation of an empty niche is not always positive, and it depends on the time the niche has been vacant and the changes in climate and the ecosystem [37], there are many cases of relatively recent extinctions at an evolutionary scale in which the occupation of their empty gap is favorable. As an example, we may mention some feral horse (*E. ferus caballus*) populations in North America [38,39], where native equid species became extinct between 10,500 and 7600 years ago [40], and thus the ecosystems may still wear some resemblance to the ones once roamed by true wild herds [41]. Some feral horses in the US (also called mustangs) originated from horses brought over from Europe by the Spanish conquistadors as early as

the XVI century, but many escaped much later, for instance, during the dust-bowl crisis in the XX century, and thus present-day feral populations have different ancestries [42]. This case is perhaps different from the feral horses in Australia (i.e., the brumbies) or in New Zealand (kaimanawa horses), where severe impacts on the native ecosystem have been documented, but where people's affection for these animals supposes a confrontation between researchers and animal rights advocates [43,44]. Additional examples include the Balearic wild goat that has filled the niche left empty by the extinct Balearic mountain goat (*Myotragus balearicus*, [16]), and thus accomplishes an ecological role. The Balearic wild goat originated from ancient domestic goats that became feral between 2300 and 2050 BC [45], and its ancient origin, along with natural selection acting during millennia, resulted in fixed ancestral characteristics. It is included in the European Mammal Assessment (EMA) [46].

An example of a lost and crucial ecological role is seed dispersal of plants that were formerly dependent on now extinct megafauna. This function has been regained with the action of livestock in some locations [47]. Mentioning a non-vertebrate animal, and in a global scenario of concern around pollinators [48], feral honeybees (*Apis mellifera*) become another interesting taxon deserving conservation [49].

In the second group of concern, we include domestic animals that have retained the only surviving genetic material of their fully extinct wild ancestors. This is the case of horses [50] in Europe, taurine cattle in Europe (*Bos taurus*), zebuine cattle in India (*B. indicus*), dromedaries (*Camelus dromedarius*) in some areas of West Asia and North Africa, and the domestic Bactrian camel (*C. bactrianus*), as it has been described as a descendant of a wild species different to the wild Bactrian camel (*C. ferus*) [51]. Both domestic camel species maintain a high number of ancestral characteristics and represent early stages of the domestication process [51], thus, they both may easily fill the niche of their ancestors. Regarding horses and cattle in Europe, they incarnate the living descendants of wild horses (*E. ferus ferus*) and aurochs (*B. primigenius*), respectively. In addition, the genetic pool of the extinct Indian auroch (*B. primigenius namadicus*) is largely present in the zebuine cattle (*B. indicus*) [52]. Furthermore, feral livestock may be a source of genetic variation with potential commercial, historical, aesthetic, or scientific value, including primitive traits or rare adaptations [53]. We must also consider some populations of disputed taxonomy. Examples are the free-living water buffaloes (*Bubalus arnee/B. bubalis*) of Sri Lanka. The most widely accepted hypothesis is that these animals originated from domestic stock. Although wild water buffalos were native to Sri Lanka, it is unlikely these populations have survived without introgression. However, these populations retain the ancestral phenotype and have cultural importance [54]. The "feral donkeys" of Jbel Elba (Egypt) are yet another case. Many consider these animals genuinely wild Nubian asses (*Equus africanus africanus*), but genetic studies must be undertaken to support this [55]. The Przewalski horse (*E. ferus przewalskii*), on the other hand, has recently been proposed as a feral descendant of early Botai's domesticated horses which escaped back to the wild about 5000 years ago [56,57]. However, this view has recently been challenged using osteological data, and Przewalski horses may indeed be a wild taxon [58]. In any case, Przewalski horses retain ancestral characteristics of truly wild horses and have been the focus of a decades-long conservation program to save them from extinction [59–61]. The interest in preserving this taxon is so high that conservationists immediately challenged the consideration of Przewalski horses as feral as soon as it was announced that these horses might not be a truly wild species [62]. This example illustrates the genetic ambiguity between feral and wild ancestors and highlights how hard it may be to distinguish between feral and wild populations.

**Table 1.** Selected feral populations of different taxa with conservation interest following different criteria: (a) Niche criterion (provision of ecosystem services), (b) extinct ancestor criterion (maintenance of unique genomic traits), and (c) cultural criterion (attribution of local or global socioeconomic value). * There is a current debate around the feral or wild origin of this taxon [62].

| Species | Common Name | Geographic Area Considered | Niche Criterion | Extinct Ancestor criterion | Cultural Criterion |
|---|---|---|---|---|---|
| *Ovis orientalis musimon* | Corsican mouflon | Corsica and Sardinia | - | - | x |
| *Capra aegagrus hircus* | Balearic Boc | Majorca | x | - | x |
| *Capra aegagrus cretica* | Kri-Kri | Crete | - | - | x |
| *Camelus dromedarius* | Dromedary | Middle East and North Africa | x | x | x |
| *Camelus bactrianus* | Camel | Central Asia | x | x | - |
| *Equus ferus caballus* | Horse | Europe and North America | x | x | - |
| *Equus ferus Przewalskii* * | Przewalski horse | Central Asia | x | x | x |
| *Bos primigenius taurus* | Taurine cattle | Europe and North-Central Asia | x | x | - |
| *Bos namadicus indicus* | Cebuine cattle | India | x | x | - |
| *Canis lupus dingo* | Dingo | Australia and South East Asia | - | - | x |
| *Canis lupus hallstromi* | New Guinea singing dog | New Guinea | - | - | x |
| *Apis mellifera* | Honey bee | Worldwide | x | - | - |

In the third category, we place the benefits provided by taxa considered cultural heritage by local communities. Here, we can include the Cyprus mouflon (*Ovis orientalis ophion*), Corsican mouflon (*O. orientalis musimon*), the Soay sheep (*O. aries*) of St. Kilda in Scotland, the Kri Kri of Crete (*C. aegagrus cretica*), the dingo (*Canis lupus dingo*) in Australia and the Guinean singing dog (*C. lupus hallstromi*) in New Guinea. All these animals have a demonstrated feral origin from domesticated breeds, but in the early stages of domestication and far back in time, generating an undisputable cultural heritage. The Cyprus mouflon descends from wild mouflons (*O. gmelini*) domesticated by peoples living in the Near East. It was introduced by Neolithic seafaring colonist to the major Mediterranean islands during their westward migration (before 7000 B.C). After Cyprus, it was introduced in Sardinia and Corsica [63]. Mouflons have been introduced in modern times in many areas of Europe for hunting purposes [64], as they are perceived as a wild sheep species, and males are sought after as trophies [65]. The origin of the Kri-Kri (or Agrimi) is also feral, as descendants from early domestic stock of goats brought to the island of Crete by the first Neolithic settlers around 8000–7500 B.C [66]. The Soay sheeps, in turn, have been described as the survivors of the earliest domestic sheep that spread through Europe in the Bronze Age [67]. This island population has been intensively monitored by successive research teams since the 1950s, and it has proved to be a model system invaluable for science in the realms of ecology and genetics (yet another asset of some feral populations).

Dingoes and New Guinea singing dogs are canids of proposed ancient feral origin in Melanesia that have elicited diverging attitudes. Dingoes are thought to have emerged around 3500–4500 years ago ([68] but see a recent assessment suggesting a much earlier feralization, [2]). In fact, a genomic study [69] suggests that dingoes are an early lineage between wolves and domestic dog breeds. Even though they are biological taxa of interest [70,71], they been vilified by interest groups, including the livestock industry in Australia, as dingoes predate on sheep. The Dingo Fence stretching over more than 5000 kms was built in the XIXth century to exclude them from sheep-producing areas, but this initiative achieved only partial success.

The term 'feral dog' is often used to denigrate this iconic animal and to promote culling measures or eradication [71]. In addition, there has been recent controversy around the status of dingoes [72], and genomic data support that, although dingoes and guinea singing dogs originated from domestic dogs in South Asia, they migrated before it was previously thought, around 8300 years ago, and that they represent a clearly differentiated canid population [2]. Some dingo populations might be introgressed by modern feral dogs, and it has been claimed that to save dingoes from extinction, it is necessary to curtail further hybridization [73]. However, recent studies demonstrate that the degree of modern dog introgression is minimal [74], and in any case, it does not affect the positive ecological role of dingoes [75].

The ongoing debate about preserving or eliminating dingoes in Australia mirrors the controversy about preserving or hunting other wild predators of domestic animals anywhere in the world. This includes wolves (*C. lupus*) and bears (*Ursus arctos*) in Europe [76,77] and North America [78], but also large wild cats, such as lions (*Panthera leo*), tigers (*P. tigris*), and leopards (*P. pardus*) in Africa and India [79,80]. Even though the above-mentioned carnivores are locally or globally endangered, and are not expected to attain large population sizes, or to become a direct threat to people's lives, some lobbies (typically hunting and farmer associations) still advocate for their eradication or oppose reintroductions or other management actions [81]. Therefore, referring to dingoes as feral dogs is no more than a form of "racism on biodiversity", namely, just a way to deprecate a predator that might potentially interfere with some human activities, even if marginally and at a negligible cost. In other words, it reflects more a prejudice than a legitimate claim of compensation for actual damages.

The case of feral horse populations deserves special attention as horses meet all proposed criteria for conservation and have a global distribution. Nonetheless, some feral populations, and particularly the largest ones in arid or semiarid areas in Australia (Table 2), may pose a risk to native vegetation due to overgrazing. We have selected some feral populations representative of all continents except Antarctica (Table 2 and Figure 2) to point out the fact that there is concern about the conservation of feral horses in practically all environments (from lowlands to highlands, wetlands, and deserts) and irrespective of their population sizes and potential impact on the environments they inhabit. This demonstrates that horse conservation is focused on the animal as a symbol, and thus conservation efforts are mostly emotion-driven while other potential reasons, such as the important functional role of horses as ecosystem engineers [82], are often secondary. Our selected examples point out, in any case, to a genuine interest in horse conservation, arising, on the one hand, from governments that protect them inside national parks and nature reserves (as with Sable Island horses in Nova Scotia, Canada, the Retuertas horse in Doñana, Spain, or the Letea Forest horse in the Danube Delta, Romania) alongside other native species, and, on the other hand, by environmental NGO's or animal rights groups. At a time when landraces of domesticated animals, the product of ancient and long-term selective processes combining the joint forces of artificial and natural selection, are receding or disappearing altogether [18], appreciating and preserving feral horses should be seen as a positive attitude. Interestingly, international institutions, such as the Food and Agriculture Organization (FAO) of the United Nations, promote the conservation of horse breeds as animal genetic resources. FAO [83] reported that numerous local horse breeds have become extinct and that about one-fourth are at risk. They also recognize emerging cultural roles of feral horses in tourism and in landscape management. In this regard, FAO lines up with rewilding efforts, which are particularly strong in Western Europe, where the human imprint is old, and landscapes are essentially anthropogenic. Rewilding with landraces of horse breeds, including, for instance, Koniks and Pottokas, is already a reality in different countries, and given the interest, many more introductions are foreseen [82]. Several studies point out the benefits of horse rewilding in abandoned wood pastures, enhancing grassland functional composition [84,85]. In fact, rewilding initiatives in Europe already count on support from the European Union, for instance, via the project "Grazelife"

(Life 18 PRE/NL002) (Grazelife.com accessed on 10 October 2021). The effect of cultural understandings on the feral horses has been, however, described by some as a distraction from effective management [86], and we strongly recommend analyzing each local case, determining the effects on each specific environment in order to take adequate measures for local ecosystem conservation.

**Table 2.** Breeds or populations of feral horses selected as examples from around the world, and organizations in charge of their management. * Brumbies are a clear example of a feral population with a negative effect on ecosystems (see text). It is important to remark that no wild equids have inhabited Australia in the past. However, there are citizen entities concerned about horses. ** There is no consensus about the feral origin of Przewalski horse, but some authors consider it a truly wild species (for more details, see main text).

| Feral Population/ Breed | Country | Estimated Population | Conservation Organization | Habitat |
|---|---|---|---|---|
| Retuertas | Spain | 200 | Doñana National Park-CSIC | Wetland |
| Sorraia | Portugal | 200 | Rewilding Europe; Sorraia.org | Riverine areas |
| Dülmen pony | Germany | 485 | Nature reserve of the Merfelder Bruch | Forest mosaic |
| Giara | Italy | 1900 | Associazione del Cavallini della Giara | Mountain plateau with wetlands |
| Exmoor pony | UK | 500 | Exmoor Pony Society | Mountain and moorlands |
| Letea Forest horse | Romania | 4000 | Danube delta Biosphere Reserve, Vier Pfoten | Wetland and forest |
| Garub wild horse | Namibia | 200 | Namibia Wild Horses Foundation | Desert |
| Gotland pony | Sweden | 5000 | Svenska Russavelsföreningen | Forest mosaic |
| Vodny island horse | Russia | 300 | Rostovsky Nature Reserve | Pastureland |
| Skyros pony | Greece | 150 | Skyrian Horse Society | Mediterranean mountains |
| Bagual | Chile and Argentina | 15,000 | Asociación de Criadores de Caballos Criollos (ACCC) | Patagonian steppes |
| Feral Andean horse | Ecuador | 97 | National Park Cotopaxi | High Andes |
| Brumby * | Australia | 300,000 | Australian Brumby Alliance (ABA) | Desert |
| Kaimanawa horse | New Zealand | 300 | New Zealand Conservation Department | Mountain forest |
| Mustang | USA | 88,000 | American Wild Horse Campaign (AWHC) | Arid mountains |
| Sable Island Horse | Canada | 500 | Sable Island National Park Reserve | Coastal dunes |
| Kaapsehoop wild horse | South Africa | 200 | Wild Horse Trust Fund | Mountain escarpments |
| Przewalski ** | Mongolia | 1900 | Foundation for the Preservation and Protection of the Przewalski Horse | Steppes |
| Kónik | Poland | >1000 | Polish Horse Breeders Association; Rewilding Europe | Pasturelands |
| Pottoka | Spain | 700 | True Nature Foundation | Mountain range |

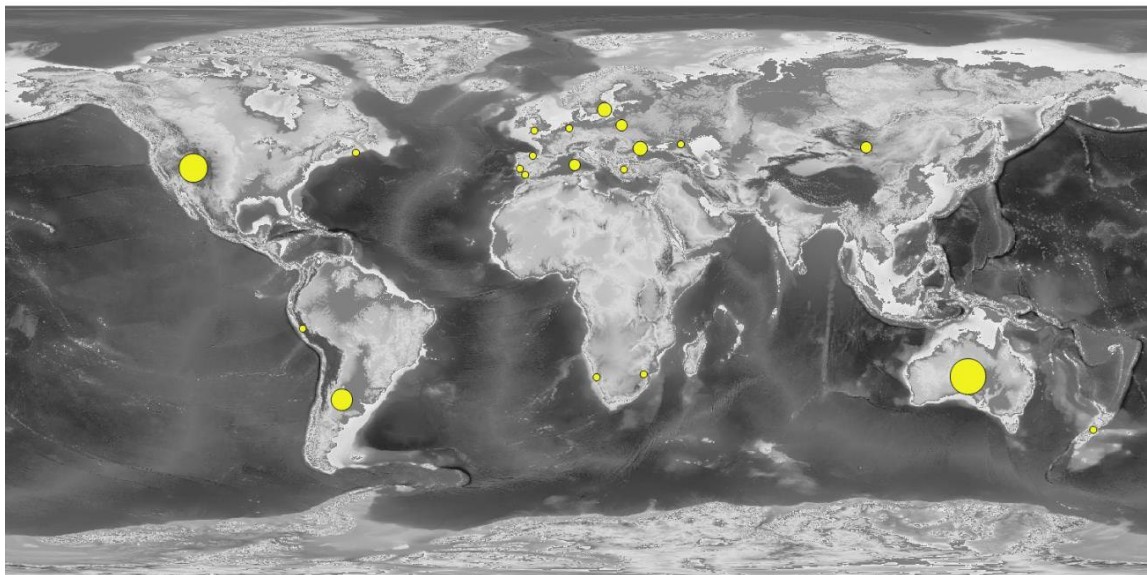

**Figure 2.** Feral horse populations provided as example in Table 2. The size of the dot is proportional to their population size. Note that the aim of this map is not to include all world populations, but a sample of the different environments used by feral horses. Background map modified from GEBCO Compilation Group (2019) GEBCO 2019 Grid (https://doi.org/10.5285/836f016a-33be-6ddc-e053-6 c86abc0788e accessed on 1 October 2021).

## 4. Conclusions

The above-mentioned taxa are just famous and notorious examples, and they may fit one or more of the proposed criteria of interest (genetic, ecological, or cultural). In addition, one species may fit some of these criteria in some areas but not in others, depending on niche availability, historic presence of their ancestor, or possible impacts. Although some feral populations may already benefit from the conservation efforts warranted by local legislation, they lack international protection and support. Practically all these taxa, except for the Przewalski horse, are excluded from IUCN assessments. It seems illogical to exclude, for example, dingoes from conservation assessments, as they represent a clearly differentiated and interesting canid taxon, as old as many glacial species or subspecies of vertebrates in the Old World [87,88]. Although most feral populations pose a clear risk to biodiversity or human activities and, for this reason, they should be managed by controlling or eradicating using ethical methods, some of them are nevertheless of conservation or cultural interest. As such, they should be awarded an equivalent status to wild animal species in order to elaborate action plans envisaging conservation measures when and where deemed necessary. We are not making proposals concerning the eradication, control or protection of the different feral species. This decision is dependent on each area and taxa, and we consider it must be analyzed in detail, being this review only a call of attention to the need for specific management revision by governments and international entities.

**Author Contributions:** E.J.R.-R. proposed the idea, wrote an early version, and selected a majority of the examples used in the text. J.G.-M. compiled the information of feral horse populations. J.J.N. wrote parts of the manuscript, revised the paper for intellectual details, and provided some of the examples. All authors have read and agreed to the published version of the manuscript.

**Funding:** This research received no external funding.

**Data Availability Statement:** Not applicable.

**Acknowledgments:** C. Ibañez, A. Redondo and I. Sánchez provided valuable comments on an early version of the manuscript.

**Conflicts of Interest:** The authors declare no conflict of interest.

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
