# Peer review of "Feral Animal Populations: Separating Threats from Opportunities"

_land, doi:10.3390/land11081370_

Round 1

Reviewer 1 Report

In their paper, Rodríguez-Rodríguez and coauthors provide a summary of case studies inherent to feralized animal population worldwide, evidencing how these are not always synonymous threats posed to local biodiversity, but also as the protagonists of unique adaption processes and sometimes the last holders of genomic features that otherwise went lost. Importantly, they highlight the cultural value which is (locally or globally) attached with some of these feralized populations, and focus on the social components which often underlie their perception in human society and, consequently, the adoption of different management approach. However, while the idea and the purpose of this review are certainly worthwhile and pertinent with the scope of Land, the MS would largely benefit from an extensive review of the English language by a native speaker. In particular, syntax of some passages is convoluted, with repetitions and wordiness. Some formal yet important aspects, such as the correct use of abbreviation and italics with scientific names, are often disregarded (this applies to the reference list too).

For reason of practicality, I have added a number of edits in the attached pdf; the authors will find a suggestion for changes or comment  by placing the cursor on the edited text.

Author Response

Thanks very much for your constructive review, we have addressed the mentioned issues as follow:

We have solved the problem with the correct use of italics and abbreviation with scientific names through all the text. Additionally, we have revised all the text in order to improve the English and implemented the changes proposed by the other reviewer. We hope this new version will be suitable for publication under adequate standards.

Reviewer 2 Report

This review has some merit and provides three useful criteria to address the merits of retention of feral species in habitats that are ancestral or outside the range of the species. It might therefore be useful to differentiate better the range issue as with the horse examples. A fourth criterion for retention of feral species, animal welfare, is addressed in the compassionate conversation literature given the typical control measure is lethal control of an identified pest. The authors might consider expanding their review to include this literature as they mention animal welfare in relation to some species like dingos. The review should be more analytical and tends towards opinion especially in the conclusions.

The manuscript has many spelling and grammatical errors, very long paragraphs and inconsistent use of italics for species names. The version I was provided to review had no line numbers until page 9 and so it would be tedious to document all these errors. I suggest a thorough proof-reading of the manuscript and a consistent style. Furthermore a more analytical approach to support the three criteria the authors raise to conserve feral animal populations would strengthen the value of the review.

Author Response

Thanks for your constructive review, we have addressed the mentioned changes as follow:

“It might therefore be useful to differentiate better the range issue as with the horse examples”

We have addressed this issue throughout the text. We have explained how each case must be studied in detail, in order to know the concrete effects on ecosystems.  As example the following paragraph, although diverse mentions are included in the text.

The effect of cultural understandings on the feral horses has been, however, described by some as a distraction from an effective management [86], and we strongly recommend to analyze each local case, determining the effects on each specific environment in order to take adequate measures for local ecosystem conservation.

“A fourth criterion for retention of feral species, animal welfare, is addressed in the compassionate conversation literature given the typical control measure is lethal control of an identified pest. The authors might consider expanding their review to include this literature as they mention animal welfare in relation to some species like dingos.”

Our aim here is not to discuss the ethic dimension once one species is considered invasive. It is mentioned because it is not possible to disregard it, but we think that animal welfare is not a criterion to consider a feral taxa of conservation interest. If a taxa is considered invasive, a threat, and not of conservation interest, then the ethical standards should arise, but this do not makes this taxa of conservation interest. We have added the following paragraph:

“It is important to remark that here, we only aim to provide an overview of feral taxa with conservation interests, differentiating from those that supposes a threat for biodiversity, and thus with not discuss the method and the animal welfare scheme that must arise once one species is considered a risk for biodiversity. However, there are a growing tendency to avoid lethal or unethical treatment in the wild animal management, supposing this a possible new criterion for the avoidance of feral populations eradication.”

“The review should be more analytical and tends towards opinion especially in the conclusions.”

We have changed some expressions in order to provide a more analytical work

“The manuscript has many spelling and grammatical errors, very long paragraphs and inconsistent use of italics for species names”

We have addressed the problem with italics as mentioned by the other reviewer as well, and revised the English of the text.

Reviewer 3 Report

The article is interesting and will make an important contribution to the field. The research shows depth, and the article addresses a broad international audience. Only few issues need to be addressed before publication.

Please separate the research goals and their importance (last sentence of introduction, starting with "We propose a detailed assessment...") from the introduction by placing them in a new paragraph. They need to be always visible. Since the goal is to produce criteria "to decide if particular feral populations should be eradicated, controlled, or protected nationally or internationally", the authors should be able to see how this goal was accomplished. It would be useful to provide in the end a table for the analyzed cases, showing whether the feral species "should be eradicated, controlled, or protected", or at least a discussion on it.

The article does not seem to fully use the "Land" template and comply with the Editing Guidelines, for example the author names are not listed in full, the header is missing on the first page, key words are separated by comma and not by semicolon, the crediting system is not the one required by the journal, references do not match the journal style and the list is numbered twice, the punctuation and capitalization of captions and figure and table references does not match the journal style, table headers use shadings, details on first page are missing from the left panel etc. Also, there are other text editing problems, for example CITES is spelled out after its second use and not after the first, line numbering is introduced only on page 11, preventing my ability to refer more flaws. However, all these issues can be accommodated at a later stage.

Author Response

The article is interesting and will make an important contribution to the field. The research shows depth, and the article addresses a broad international audience. Only few issues need to be addressed before publication.

Thanks very much for your constructive review, we have addressed the mentioned changed as follows:

Please separate the research goals and their importance (last sentence of introduction, starting with "We propose a detailed assessment...") from the introduction by placing them in a new paragraph. They need to be always visible. Since the goal is to produce criteria "to decide if particular feral populations should be eradicated, controlled, or protected nationally or internationally", the authors should be able to see how this goal was accomplished. It would be useful to provide in the end a table for the analyzed cases, showing whether the feral species "should be eradicated, controlled, or protected", or at least a discussion on it.

We have separated the paragraph. Additionally, we have discussed at the end the objectives, but without introducing a table with decisions about eradication, control or protection. This decision is dependent on each area and taxa, and we consider must be analysed in detail, being this review only a call of attention about the need of this specific management decision.

“We do not introduce proposals about eradication, control or protection of the different mentioned feral species o populations. This decision is dependent on each area and taxa, and we consider must be analysed in detail, being this review only a call of attention about the need of this specific management revision by governments and international entities”

The article does not seem to fully use the "Land" template and comply with the Editing Guidelines, for example the author names are not listed in full, the header is missing on the first page, key words are separated by comma and not by semicolon, the crediting system is not the one required by the journal, references do not match the journal style and the list is numbered twice, the punctuation and capitalization of captions and figure and table references does not match the journal style, table headers use shadings, details on first page are missing from the left panel etc. Also, there are other text editing problems, for example CITES is spelled out after its second use and not after the first, line numbering is introduced only on page 11, preventing my ability to refer more flaws. However, all these issues can be accommodated at a later stage.

Thanks very much for pointing out these flats. We have tried to solve many of the mentioned issues, although we have some problems with the template respect numeration etc. We hope these flaws can be solved upon acceptance.

Round 2

Reviewer 2 Report

Authors have made an adequate response to my comments. They need to review inserted text for minor errors which can be addressed in proofs.

Author Response

Many thanks for your constructive review. We have revised the mentioned issues